# Predictive ability of plasma concentration of triglyceride/high density lipoprotein-cholesterol ratio for cardiometabolic variables in a sub-Sahara black African adolescent population–Nigerians

Susan J. Holdbrooke[1‡], Bamgboye M. Afolabi[1,2‡*]

1 Department of Biochemistry and Nutrition, Nigerian Institute of Medical Research, Lagos, Nigeria,
2 Health, Environment and Development Foundation, Lagos, Nigeria

‡ SJH and BMA are joint first authors.
* bmafolabi@gmail.com

## Abstract

### Introduction

This study aimed to determine whether triglyceride/high-density lipoprotein cholesterol (TG/HDL-c) ratio, which has been shown to be an innovative proxy marker of atherogenic indicator in the human plasma, and an indicator of the metabolic syndrome (MetS) and insulin resistance (IR), can predict systolic hypertension (SHT), diastolic hypertension (DHT), diabetic fasting plasma glucose (dFPG), hypertriglyceridemia (HTG), hypercholesterolemia (HCHOL), low high-density lipoprotein-cholesterol (HDL-c) and high low-density cholesterol (LDL-c) in the Nigerian adolescent population living in metropolitan Lagos, Southwest Nigeria.

### Methods

A dietary and nutritional survey (DNS) was conducted from October 2019 to March 2020. A total of 650 adolescent participants were recruited using a four-stage stratified sampling method but statistical analysis was restricted to the 613 who had complete anthropometric and clinical data. The sensitivity, specificity, and distance to the corner on the receiver operating characteristic (ROC) curve in each TG/HDL level were calculated. The shortest distance in the ROC curves was used to determine the optimal cutoff of the TG/HDL-c ratio for detecting MetS, Systolic and diastolic hypertension and diabetes.

### Results

In all the subjects, the median TG/HDL-c ratio was significantly higher (P-value <000001) only among those with, than those without MetS (8.2 vs 3.0), those with

**Data availability statement:** All relevant data and code have been uploaded to Figshare at DOI https://doi.org/10.6084/m9.figshare.29618150.

**Funding:** The author(s) received no specific funding for this work.

**Competing interests:** The authors have declared that no competing interests exist.

and without hypertriglyceridemia (4.2 vs 1.3), hypercholesterolemia (5.3 vs 2.8), low high-density lipoprotein-cholesterol (7.2 vs 2.5) and high low-density lipoprotein-cholesterol (3.3 vs 1.8, P-value = 0.0001). The prevalence of systolic hypertension, diabetic FPG, total cholesterol and LDL-c was higher with higher TG/HDL-c ratio while that of HDL-c was lower with higher TG/HDL-c in all subjects. TG/HDL-c ratio was strong in predicting dyslipidemia, especially hypertriglyceridemia and hypercholesterolemia in early and mid-adolescents and cardiometabolic risk factors of two or more in adolescents. An inverse relationship was observed between the stages of adolescence and cutoffs for MetS.

## Conclusions

TG/HDL-c ratio effectively predicted MetS, hypertriglyceridemia and hypercholesterolemia among early and mid-adolescent Nigerians. As a tool, the TG/HDL-c ratio should be considered for the initial prediction of MetS and some of its dyslipidimic components. Further studies are needed to confirm findings in this study.

## Introduction

The aging progression and metabolism in humans are intimately coupled, and various age-related alterations in body components, such as additional central adiposity and sarcopenia, have their foundations in fundamental aging processes, further exacerbated by a sedentary lifestyle [1]. Dysregulated metabolism has been identified as one of the hallmarks of aging mechanisms [2,3]. A study remarks that, due to the aging process, the elderly are at a higher risk of developing intra-abdominal obesity and metabolic syndrome [4]. A study reported the prevalence of MetS in an elderly population as varying from 11% to 43% (median 21%) according to the WHO, and 23% to 55% (median 31%) according to NCEP. Another study pronounced that MetS in the elderly is a verified risk factor for cardiovascular (CV) morbidity, especially stroke and coronary heart disease (CHD), and mortality [5] But what is the picture in pre-adulthood period of life, especially in adolescence and most especially among a Black African adolescent population? There is a consensus that MetS consists of obesity, insulin resistance, dyslipidemia, and hypertension [6,7] leading to increased risk of cardiovascular disease (CVD) and renal events [8]. The foremost cause of morbidity and mortality that impose substantial social and economic burden globally, are cardiovascular diseases (CVDs) [9]. Tong et al concluded that the burden of CVD in adolescent and young adults is a significant global health challenge [10]. Furthermore, Sun et al submitted that the countries/territories with low and low-middle SDI had a higher burden of CVDs than the countries/territories with high and high-middle socio-demographic index and concluded that there was (and probably still is) a substantial global burden of CVDs in youths and young adults in 2019 [9]. A systematic review and meta-analysis shows an upward trend in the population-based prevalence of CHD in Africa [11]. In urban Nigeria, there has been a rising prevalence of hypertension

[12]. A 150% increase in the prevalence of cardiovascular disease has also been reported [13]. Hypertension affects up to 46% of Nigerian adults and a rising proportion of Nigerian adolescents [14,15]. Major cooking oils in Nigeria are palm oil and groundnut oil, which are locally produced and sold at markets. Because these low-cost oils are widely available, people consume them regularly, which contributes to the rise in obesity, metabolic syndrome, and type 2 diabetes [16,17] thus associating food consumption to the development of MetS and invariably CVD. A study has reported dietary intake among adolescents with MetS, with the conclusion that high weekly consumption of animal protein appears to be a risk factor for the development of MetS, especially among boys [18]. These is ample evidence to suggest that serum lipid levels are rising in Nigeria. For example, in 2010, one study reported that the lipid profiles of adolescent Nigerians were low but gradually rising towards values seen in places where coronary heart disease occurs in epidemic proportions, indicating an increased risk of arteriosclerosis in future generations [19]. Later in 2015, Yar'adua et al reported a low risk for cardiovascular disease among Nigerian adolescents [20]. However, in 2021, Onyiriuka et al reported dyslipidaemia in a high proportion of adolescents with either overweight/obesity or normal BMI, with the former having a higher proportion and that low HDL-c level was the most common abnormality in dyslipidemia [21]. Although the majority of clinical CVD events occur at middle and older ages, substantial evidence indicates that atherosclerosis has its origins in childhood [22]. Over a century ago, Klotz and Manning [23] in autopsy findings, reported fatty streaks in large arteries of 6-year-old children. Of recent, intimal lesions were also observed in all aortas and in more than half of the right coronary arteries of adolescents aged 15–19 years [24]. Thus, without effective intervention in adolescence, atherosclerosis-related diseases may soar in the near future in Nigeria because of abnormal lipid metabolism which has been confirmed as an integral aspect of atherosclerotic pathogenesis, essential to the occurrence of CVD. Initially proposed by Gaziano et al, TG/HDL-c ratio refers to an atherogenic index which has been confirmed as a major independent predictor of cardiovascular disease [25]. Kannel et al [26] and Bertoluci et al [27] associated a high TG/HDL-c ratio to the risk of cardiovascular events. TG/HDL-C ratio has been shown to be a more dependable substitue for forecasting MetS and cardiometabolic risks compared to other lipid ratios such as TC/HDL-C, LDL-C/HDL-C and non-HDL-C/HDL-C with a high predictive value [28–30]. Ample examples exist of elevated cardiometabolic risk for TG/HDL-C ratios between 2 and 3 among children and adolescents, with sensitivity and specificity similar to the criteria recently suggested to diagnose MetS [31–33]. The current literature contains a plethora of trials and studies that established the clinical usefulness of TG/HDL-C ratio in distinguishing between children and adolescents with MetS [34], predicting coronary artery disease (CAD), peripheal artery disase (PAD) and crebrovascular accident (CVA) in pediatric and adolescent MetS [35] Few studies in Black Africa have focused CVD on adults and fewer still in adolescents. Yet, the link between TG/HDL-c ratio and cardiovascular risk factors among adolescent Black African, especially Nigerians, remains unknown. Therefore, the objective of this study was to investigate the relationship between TG/HDL-c ratio and cardiovascular risk factors among this group of people. The study also attempted to evaluate the optimal cutoff points of TG/HDL-c ratio to predict cardiovascular risk factors in Nigerian early, mid- and late adolescents.

## Materials and methods

### Sample design

Detailed description of the study population and the methods used in conducting the study have already been extensively described in a previous publication [18]. However, a brief clarification is rendered below. The study, descriptive and cross-sectional, was conducted at Biochemistry and Nutrition Department of the Nigerian Institute of Medical Research (NIMR), Lagos, Nigeria. In the study, eligible adolescents aged 10–19 years, were selected from different secondary schools for a dietary and nutrition survey (DNS) and primary data were collected from them. Originally, 624 participants (241 boys and 383 girls) were recruited but the data of only 613 (98.2%) were processed and analyzed in this article because of incomplete questionnaire responses. On the 4th of February 2019, the Institutional Review Board of the

Nigerian Institute of Medical Research (NIMR IRB (IRB/18/062) approved the study protocol, and the study was carried out in accordance with the Declaration of Helsinki (2000). Recruitment of participants into the study started on Thursday, October 10, 2019 (10/10/19) and ended on Friday, March 20, 2020 (03/20/20). The purpose of the study was clearly explained to each participating student. Each student was given an ample opportunity to ask questions for better understanding of the study. Thereafter, students who, in the presence of their teachers, gave verbal assent to participate in the study, were then given consent forms for their parents to also read, understand and sign or thumbprint.

**Study site.** This study was conducted in metropolitan Lagos, Lagos State, southwest Nigeria, a city-state, an economically viable and most populous city in the country. When this study was conducted, there were 616 registered public (372, 60.4%) and private (244, 39.6%) secondary schools in the local government areas (LGAs) selected for the study [36].

**Study population.** These were students aged 10–19 year who were attending government-authorized secondary schools in Lagos State Nigeria. The age range between 10–19 years had been defined by the World Health Organization (WHO) as the period of adolescence [37].

**Sample size.** The sample size, sampling technique and procedure for the study have also been reported in an earlier publication [38]. Simply stated, the sample size was determined for a specific population with 95% confidence interval, 54% proportion, a margin of error 5%, and accepting 12% non-response. The sample size was set at 650 students to guarantee that results of the study are expressive of majority of Nigerian ethnic groups inhabitant in Lagos State, and to cater for attrition and lost data. Flowchart of participants' recruitment into the study is illustrated in Fig 1.

**Eligibility criteria.** To be eligible, participants must be 10–19 years of age, dark-skinned and indigenous citizen of Nigeria. He or she must be resident in the community for a minimum of two years in the respective area of the study, must also be a regular student at any of the selected secondary schools. The State Ministry of Education and parents must give their documented approvals for the study and each student must also be willing to participate. Students who were

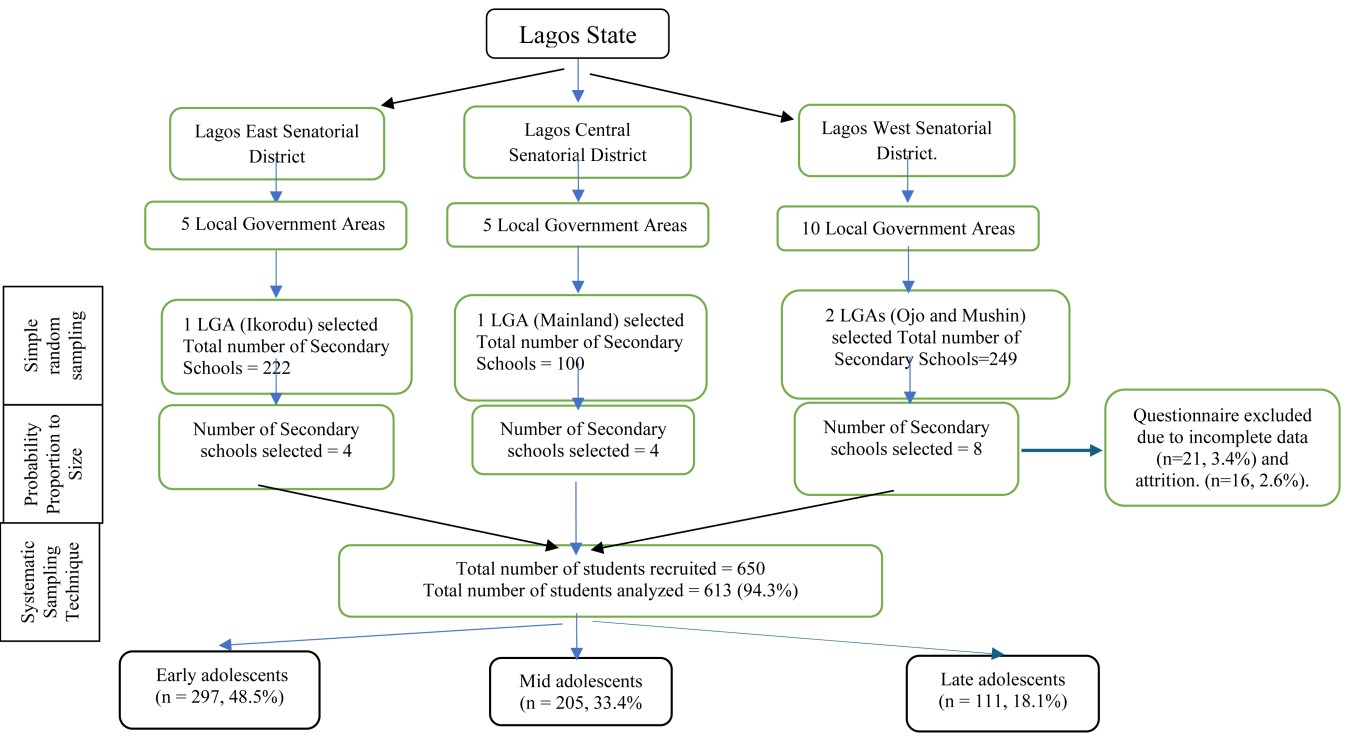

**Fig 1. Flowchart of recruitment of study participants.**

(i) on therapeutic diet or drugs (ii) on admissions to a health facility in previous 6 months, (iii) known diabetics (iv) having vascular/liver/renal or any other chronic illness and those taking lipid-lowering medications were excluded. Also excluded were females who were (i) pregnant (ii) suspected to be pregnant (iii) breastfeeding and (vi) those on oral contraceptives.

*Questionnaire:* A semi-structured questionnaire was administered to record socio-demographic, dietary and nutritional data and to record anthropometric measurements and readings of systolic and diastolic blood pressure. These and all other laboratory procedures have earlier been documented [38].

*Statistical analysis:* In this report, participants were segregated by stage of adolescence (early or 10–13 years of age, mid or 14–17 years of age and late or 18–19 years of age). The data was subjected to descriptive statistics, Analysis of variance and ROC analysis using NCSS version 22 (Kaysville, Utah, USA). The sensitivity and specificity of each TG/HDL level for the detection of MetS, systolic and diastolic hypertension, diabetes, hypertriglyceridemia, hypercholesterolemia, low high-density lipoprotein cholesterol, high low-density lipoprotein cholesterol and ≥2 of these risk factors were calculated by creating dichotomous cardiometabolic variables for each TG/HDL-c value. Furthermore, the cutoff point, distance on the receiver operating characteristic (ROC) curve, sensitivity and specificity of each TG/HDL value was calculated using the NCSS statistical software. The TG/HDL-c value with the shortest distance on the ROC curve was considered in the determination of optimal cutoff for the identification of any of the cardiometabolic risk factors. The overall performance of the TG/HDL-c test for distinguishing cardiovascular risk factors was rated by calculating the area under the curves (AUC) in which an AUC of 1 was regarded to hold a flawless discriminatory power, and an AUC of 0.5 or below to submit that the discriminatory power is just accidental. Values were reported as mean (±), standard deviation (SD) and 95% CI for the continuous variables. Shapiro-Wilk's Normality test for normality of data distribution for continuous measures was conducted and when the test failed (i.e., $p$-value was < 0.05, the normality of data distribution was rejected, and Kruskal-Wallis ANOVA was used to determine differences between 2 or more medians. An unadjusted p-value <0.05 was considered statistically significant. Data were illustrated as tables, graphs, charts, and figures.

**Definitions.** This has also been earlier documented [38]. Briefly, unacceptable lipid levels were classified as (i) total cholesterol ≥200 mg/dL (ii) low-density lipoprotein-cholesterol (LDL-c) ≥ 130 mg/dL (iii) triglycerides (TG) ≥130 mg/dL (hypertriglyceridemia), (iv) high-density lipoprotein cholesterol (HDL-c) < 40 mg/dL [39]. (v) Fasting plasma glucose (FPG) of ≥126 mg/dL was taken as diabetic; (vi) stage 1 hypertension, BP 130–139/80–89, and stage 2 ≥ 140/90 mmHg [40]. Triglyceride/High-density lipoprotein-c (TG/HDL-c) is the ratio of triglyceride divided by high-density lipoprotein-c of each study participant.

## Results

Socio demographic and anthropometric characteristics of the study subjects, measurements, laboratory analyses as well as the prevalence of metabolic syndrome, have already been documented extensively in previous publications [41]. As presented in Table 1, the overall median TG/HDL-c ratio was significantly higher (P-value=<0.00001) in adolescents with (n = 52, 8.5%; median = 8.2), compared to those without (n = 561, 91.5%: median = 3.0), MetS. This significance is also evident in each stage of adolescence – early, mid and late, of the study subjects. On the other hand, overall median TG/HDL-c ratio among all subjects with either systolic or diastolic hypertension and those with diabetic fasting plasma glucose were evenly distributed, regardless of their stage of adolescence. Table 2 highlights age-standardized cardiovascular disease risk factors – SBP, DBP, FPG, TG, TC, HDL-c and LDL-c, relative to TG/HDL-c category in all the study subjects and in different stages of adolescence. With an increasing TG/HDL-c, plasma total cholesterol increased from 27.2 mg/dL among those with TG/HDL-c ≤ 1.0 to 223.4 mg/dL in those with TG/HDL-c > 5.0, plasma HDL-c decreased from 123.6 mg/dL to 48.6 mg/dL, though these changes did not reach any level of significance. In early adolescence, plasma TC increased with an increase in TG/HDL-c ratio from 36.1 among subjects with TG/HDL-c ratio of ≤1.0 to 213.8 mg/dL in those with TG/HDL-c > 5.0 while plasma HDL-c decreased from 117.0 to 49.1 mg/dL. Table 3 illustrates age-standardized prevalence of risk factors by TG/HDL-c category. Table 3 illustrates the age-specific prevalence of risk factors by TG/HDL-c ratio among the study subjects. Overall, the prevalence of hypertriglyceridemia and hypercholesterolemia significantly increased

Table 1. Median values of TG/HDL-c ratio relative to cardiometabolic risk factors in Nigeria adolescents.

| Variables | Metabolic syndrome | | Cardiometabolic risk factors | | | | | | | | | | | | | |
|---|---|---|---|---|---|---|---|---|---|---|---|---|---|---|---|---|
| | | | Hypertension | | | | Diabetic FPG | | Hypertriglyceridemia | | Hypercholesterolemia | | Low HDL-c | | High LDL=c | |
| | | | Systolic | | Diastolic | | | | | | | | | | | |
| | Yes | No | Yes | No | Yes | No | Yes | No | Yes | No | Yes | No | Yes | No | Yes | No |
| All (n=613) | (n=52, 8.5%) | (n=561, 91.5%) | (n=109, 17.8%) | (n=504 82.2%) | (n=8, 1.3%) | (n=605, 98.7%) | (n=71, 11.6%) | (n=542, 88.4%) | (n=431, 70.3%) | (n=182, 29.7%) | (n=262, 42.7%) | (n=351, 57.3%) | (n=153, 25.0%) | (n=460, 75.0%) | (n=525, 85.6%) | (n=88, 14.4%) |
| | 8.2 | 3.0 | 3.3 | 3.2 | 3.1 | 3.2 | 3.1 | 3.3 | 4.2 | 1.3 | 5.3 | 2.8 | 7.3 | 2.5 | 3.3 | 1.8 |
| | 52.2 (<0.00001)* | | 0.0001 (>0.05) | | 0.05 (>0.05) | | 1.63 (>0.05) | | 200.7 (<0.00001) | | 119.8 (<0.00001) | | 58.8 (<0.00001) | | 14.7 (0.0001) | |
| Early adolescents | (n=24, 46.2%) | (n=273, 48.7%) | (n=52, 49.5%) | (n=245, 48.6%) | (n=4, 50.0%) | (n=293, 48.4%) | (n=44, %) | (n=253, %) | (n=201, 46.6%) | (n=96, 52.8%) | (n=112, 42.7%) | (n=185, 52.7%) | (n=70, 45.7%) | (n=227, 49.4%) | (n=257, 49.0%) | (n=40, 45.5%) |
| | 6.4 | 2.9 | 2.6 | 3.1 | 1.3 | 3.0 | 3.0 | 3.0 | 3.9 | 1.2 | 4.4 | 2.1 | 6.7 | 2.4 | 3.1 | 1.8 |
| | 16.3 (<0.00001)* | | 0.77 (>0.05) | | 0.90 (>0.05) | | 0.27 (>0.05) | | 108.1 (<0.00001) | | 51.2 (<0.00001) | | 24.8 (<0.00001) | | 12.5 (0.0004) | |
| Mid-adolescents | (n=21, 40.4%) | (n=184, 32.8%) | (n=40, 36.7%) | (n=165 32.7%) | (n=2, 25.0%) | (n=203, 33.6%) | (n=16, %) | (n=189, %) | (n=150, 34.8%) | (n=55, 30.2%) | (n=99, 37.8%) | (n=106, 30.2%) | (n=54, 35.3%) | (n=151, 32.8%) | (n=174, 33.1%) | (n=31, 35.2%) |
| | 9.4 | 3.3 | 3.9 | 3.7 | 3.3 | 3.8 | 3.5 | 3.7 | 4.9 | 1.3 | 6.2 | 1.9 | 7.7 | 2.7 | 3.8 | 2.7 |
| | 28.6 (<0.00001)* | | 0.41 (>0.05) | | 0.06 (>0.05) | | 0.47 (>0.05) | | 75.5 (<0.00001) | | 47.8 (<0.00001) | | 23.0 (<0.00001) | | 2.21 (0.14) | |
| Late adolescents | (n=7, 13.5%) | (n=104, 18.5%) | (n=17, 15.6%) | (n=94, 18.7%) | (n=2, 25.0%) | (n=109, 18.0%) | (n=11, %) | (n=100, %) | (n=80, 18.6%) | (n=31, 17.0%) | (n=51, 19.5%) | (n=60, 17.1%) | (n=29, 19.0%) | (n=82, 17.8%) | (n=94, 17.9%) | (n=17, 19.3%) |
| | 11.3 | 3.2 | 3.3 | 3.4 | 6.1 | 3.3 | 3.1 | 3.4 | 4.2 | 1.2 | 5.5 | 1.9 | 9.8 | 2.6 | 3.4 | 1.8 |
| | 8.2 (<0.00001)* | | 0.01 (>0.05) | | 0.67 (>0.05) | | 0.41 (>0.05) | | 19.4 (0.00001) | | 19.7 (0.00001) | | 10.3 (0.001) | | 1.87 (0.17) | |

*Kruskal-Wallis ANOVA (P-value); H0=All medians are equal; H1=At least two medians are different. Yellow highlight=Reject H0

**Table 2. Age-standardized cardiovascular disease risk factors relative to TG/HDL-c category in different stages of adolescence in Nigeria.**

| Stages of adolescence | Cardiovascular risk factors | TG/HDL-c ratio category | | | | | | P-value |
|---|---|---|---|---|---|---|---|---|
| | | ≤1.0 | >1.0 to ≤2.0 | >2.0 to ≤3.0 | >3.0 to ≤4.0 | >5.0 to ≤5.0 | >5.0 | |
| All (n=613) | Population (%) | 16 (2.6) | 51 (8.3) | 75 (12.2) | 51 (8.3) | 48 (7.8) | 372 (60.7) | – |
| | Age (yrs) | 14.2 (2.5) | 14.7 (2.1) | 14.7 (2.0) | 14.5 (2.2) | 14.4 (1.9) | 14.8 (2.2) | 0.10 |
| | Systolic blood pressure (SBP) mm Hg | 101.5 (11.4) | 107.9 (12.0) | 109.2 (11.1) | 103.4 (13.3) | 106.7 (13.6) | 109.2 (12.3) | 0.33 |
| | Diastolic blood pressure (DBP) mm Hg | 58.6 (8.4) | 64.7 (9.4) | 66.2 (9.2) | 64.4 (8.5) | 65.6 (11.2) | 67.0 (9.6) | 0.0008 |
| | Fasting plasma glucose (FPG) mg/dL | 67.4 | 82.4 | 88.1 | 88.1 | 81.4 | 89.3 | 0.37* |
| | Triglyceride (TG) mg/dL | 155.0 | 194.5 | 217.7 | 182.5 | 193.4 | 204.2 | 0.14* |
| | Total cholesterol (TC) mg/dL | 27.2 | 77.6 | 105.5 | 139.7 | 154.0 | 223.4 | 0.14* |
| | High-density lipoprotein cholesterol (HDL-c) mg/dL | 123.6 | 106.0 | 80.4 | 79.0 | 70.4 | 48.6 | 0.06* |
| | Low-density lipoprotein cholesterol (LDL-c) mg/dL | 370.8 | 209.7 | 275.3 | 291.1 | 310.3 | 291.1 | 0.26* |
| Metabolic syndrome (Freq. (%)) | | 0 (0.0) | 1 (2.0) | 0 (0.0) | 2 (3.9) | 0 (0.0) | 49 (13.2) | – |
| Early adolescents (n=297) | Population (%) | 9 (3.0) | 25 (8.4) | 35 (11.8) | 29 (9.8) | 28 (9.4) | 171 (57.6) | – |
| | Age (yrs) | 12.4 (1.5) | 13.0 (1.2) | 13.0 (1.1) | 13.0 (1.5) | 13.2 (1.3) | 12.9 (1.3) | 0.13 |
| | Systolic blood pressure (SBP) mm Hg | 104.6 (12.2) | 106.7 (11.3) | 105.4 (11.0) | 103.4 (12.4) | 104.1 (11.4) | 108.1 (13.0) | 0.79 |
| | Diastolic blood pressure (DBP) mm Hg | 59.2 (7.2) | 65.4 (8.5) | 64.3 (8.8) | 65.3 (8.6) | 62.6 (9.2) | 66.7 (9.8) | 0.0008 |
| | Fasting plasma glucose (FPG) mg/dL | 72.1 | 91.2 | 72.9 | 85.7 | 79.6 | 90.9 | 0.46* |
| | Triglyceride (TG) mg/dL | 156.0 | 189.4 | 223.9 | 209.0 | 212.3 | 208.0 | 0.49* |
| | Total cholesterol (TC) mg/dL | 36.1 | 77.6 | 104.3 | 133.1 | 166.5 | 213.8 | 0.34* |
| | High-density lipoprotein cholesterol (HDL-c) mg/dL | 117.0 | 106.0 | 84.5 | 74.5 | 71.3 | 49.1 | 0.22* |
| | Low-density lipoprotein cholesterol (LDL-c) mg/dL | 359.9 | 135.2 | 291.1 | 300.3 | 331.9 | 295.1 | 0.39* |
| Metabolic syndrome (Freq. (%)) | | 0 (0.0) | 1 (1.6) | 0 (0.0) | 2 (3.7) | 0 (0.0) | 21 (30.4) | – |
| Mid-adolescents (n=205) | Population (%) | 4 (25.0) | 16 (31.4) | 27 (36.0) | 14 (27.5) | 14 (29.2) | 130 (34.9) | – |
| | Age (yrs) | 15.9 (0.6) | 15.7 (0.5) | 15.6 (0.5) | 15.7 (0.5) | 15.2 (0.4) | 15.6 (0.6) | 0.20 |
| | Systolic blood pressure (SBP) mm Hg | 96.7 (9.7) | 110.7 (15.1) | 112.6 (8.7) | 105.1 (14.8) | 108.0 (16.2) | 109.8 (12.4) | 0.45 |
| | Diastolic blood pressure (DBP) mm Hg | 57.0 (10.1) | 65.4 (12.2) | 69.2 (8.7) | 62.7 (9.5) | 68.0 (11.3) | 67.3 (10.1) | 0.25 |
| | Fasting plasma glucose (FPG) mg/dL | 27.2 | 79.1 | 93.4 | 85.2 | 79.1 | 86.9 | 0.53* |
| | Triglyceride (TG) mg/dL | 133.0 | 199.1 | 212.8 | 177.8 | 186.8 | 199.0 | 0.47* |
| | Total cholesterol (TC) mg/dL | 65.6 | 81.0 | 108.9.4 | 136.0 | 128.2 | 231.6 | 0.31* |
| | High-density lipoprotein cholesterol (HDL-c) mg/dL | 238.2 | 109.0 | 79.8 | 80.5 | 62.4 | 47.1 | 0.30* |
| | Low-density lipoprotein cholesterol (LDL-c) mg/dL | 343.1 | 244.5 | 275.3 | 305.8 | 290.7 | 291.4 | 0.39* |
| Metabolic syndrome (Freq. (%)) | | 0 (0.0) | 0 (0.0) | 0 (0.0) | 1 (4.8) | 3 (15.0) | 17 (22.4) | – |
| Late adolescents (n=111) | Population (%) | 3 (18.7) | 10 (19.6) | 13 (17.3) | 8 (15.7) | 6 (12.5) | 71 (19.1) | – |
| | Age (yrs) | 17.4 (0.5) | 17.7 (0.6) | 17.7 (0.8) | 17.9 (0.7) | 17.8 (0.8) | 17.8 (0.7) | 0.40 |
| | Systolic blood pressure (SBP) mm Hg | 98.7 (11.5) | 106.6 (7.2) | 112.0 (13.4) | 100.2 (15.3) | 116.0 (14.7) | 110.9 (10.4) | 0.63 |
| | Diastolic blood pressure (DBP) mm Hg | 58.7 (12.9) | 61.9 (6.5) | 64.9 (10.0) | 64.4 (7.0) | 74.0 (15.3) | 67.4 (7.9) | 0.92 |
| | Fasting plasma glucose (FPG) mg/dL | 63.2 | 73.5 | 88.1 | 90.0 | 100.6 | 87.2 | 0.46* |
| | Triglyceride (TG) mg/dL | 169.4 | 164.6 | 164.2 | 178.3 | 152.9 | 203.2 | 0.41* |
| | Total cholesterol (TC) mg/dL | 13.9 | 65.4 | 103.0 | 176.8 | 170.9 | 233.9 | 0.38* |
| | High-density lipoprotein cholesterol (HDL-c) mg/dL | 67.6 | 68.3 | 80.4 | 101.7 | 76.9 | 48.0 | 0.35* |
| | Low-density lipoprotein cholesterol (LDL-c) mg/dL | 393.5 | 263.9 | 207.4 | 212.0 | 303.2 | 265.5 | 0.45* |
| Metabolic syndrome (Freq. (%)) | | 0 (0.0) | 0 (0.0) | 0 (0.0) | 1 (5.0) | 1 (9.1) | 5 (14.7) | – |

*Kruskal-Wallis ANOVA.

**Table 3. Age-specific prevalence of risk factors by TG/HDL-c category among Nigerian adolescents.**

| Stages of adolescence | Cardiovascular risk factors | TG/HDL-c ratio category | | | | | | P-svalue |
|---|---|---|---|---|---|---|---|---|
| | | ≤1.0 | >1.0 to ≤2.0 | >2.0 to ≤3.0 | >3.0 to ≤4.0 | >5.0 to ≤5.0 | >5.0 | |
| All (n=613) | Population (%) | 16 (2.6) | 51 (8.3) | 75 (12.2) | 51 (8.3) | 48 (7.8) | 372 (60.7) | – |
| | Systolic hypertension | 3 (18.7) | 8 (15.7) | 13 (17.3) | 9 (17.6) | 9 (18.7) | 67 (18.0) | 1.00 |
| | Diastolic hypertension | 0 (0.0) | 0 (0.0) | 3 (4.0) | 0 (0.0) | 0 (0.0) | 5 (1.3) | 0.27 |
| | Diabetic FPG | 1 (6.2) | 12 (23.5) | 6 (8.0) | 7 (13.7) | 6 (12.5) | 39 (10.5) | 0.10 |
| | Hypertriglyceridemia | 0 (0.0) | 8 (15.7) | 19 (25.3) | 31 (60.8) | 30 (62.5) | 343 (92.2) | <0.00001 |
| | Hypercholesterolemia | 0 (0.0) | 0 (0.0) | 5 (6.7) | 11 (21.6) | 16 (33.3) | 230 (61.8) | <0.00001 |
| | Low HDL | 0 (0.0) | 3 (5.9) | 2 (2.7) | 4 (7.8) | 5 (10.4) | 139 (37.4) | <0.00001 |
| | High LDL | 11 (68.7) | 33 (64.7) | 62 (82.7) | 40 (78.4) | 46 (95.8) | 333 (89.5) | <0.00001 |
| Early adolescents (n=297) | Population (%) | 9 (3.0) | 25 (8.4) | 35 (11.8) | 29 (9.8) | 28 (9.4) | 171 (57.6) | – |
| | Systolic hypertension | 2 (22.2) | 4 (16.0) | 7 (20.0) | 6 (20.7) | 6 (21.4) | 27 (15.8) | 0.95 |
| | Diastolic hypertension | 0 (0.0) | 0 (0.0) | 3 (8.6) | 0 (0.0) | 0 (0.0) | 1 (0.6) | 0.008 |
| | Diabetic FPG | 0 (0.0) | 7 (28.0) | 4 (11.4) | 4 (13.8) | 5 (17.9) | 24 (14.0) | 0.34 |
| | Hypertriglyceridemia | 0 (0.0) | 2 (8.0) | 7 (20.0) | 17 (58.6) | 18 (64.3) | 157 (91.8) | <0.00001 |
| | Hypercholesterolemia | 0 (0.0) | 0 (0.0) | 1 (2.9) | 6 (20.7) | 10 (35.7) | 95 (55.6) | <0.00001 |
| | Low HDL | 0 (0.0) | 3 (12.0) | 1 (2.9) | 2 (6.9) | 3 (10.7) | 61 (35.7) | <0.00001 |
| | High LDL | 6 (66.7) | 13 (52.0) | 31 (88.6) | 23 (79.3) | 28 (100.0) | 156 (91.2) | <0.00001 |
| Mid-adolescents (n=205) | Population (%) | 4 (25.0) | 16 (31.4) | 27 (36.0) | 14 (27.5) | 14 (29.2) | 130 (34.9) | – |
| | Systolic hypertension | 1 (25.0) | 4 (25.0) | 5 (18.5) | 0 (0.0) | 1 (7.1) | 29 (22.3) | 0.33 |
| | Diastolic hypertension | 0 (0.0) | 0 (0.0) | 0 (0.0) | 0 (0.0) | 0 (0.0) | 2 (1.5) | 0.95 |
| | Diabetic FPG | 0 (0.0) | 3 (18.7) | 2 (7.4) | 2 (14.3) | 0 (0.0) | 9 (6.9) | 0.40 |
| | Hypertriglyceridemia | 0 (0.0) | 3 (18.7) | 8 (29.6) | 8 (57.1) | 6 (42.9) | 125 (96.2) | <0.00001 |
| | Hypercholesterolemia | 0 (0.0) | 0 (0.0) | 3 (11.1) | 2 (14.3) | 4 (28.6) | 90 (69.2) | <0.00001 |
| | Low HDL | 0 (0.0) | 0 (0.0) | 0 (0.0) | 1 (7.1) | 2 (14.3) | 51 (39.2) | <0.00001 |
| | High LDL | 2 (50.0) | 13 (81.2) | 21 (77.8) | 12 (85.7) | 12 (85.7) | 114 (87.7) | 0.32 |
| Late adolescents (n=111) | Population (%) | 3 (18.7) | 10 (19.6) | 13 (17.3) | 8 (15.7) | 6 (12.5) | 71 (19.1) | – |
| | Systolic hypertension | 0 (0.0) | 0 (0.0) | 1 (7.7) | 3 (37.5) | 2 (33.3) | 11 (15.5) | 0.19 |
| | Diastolic hypertension | 0 (0.0) | 0 (0.0) | 0 (0.0) | 0 (0.0) | 0 (0.0) | 2 (5.9) | 0.95 |
| | Diabetic FPG | 1 (33.3) | 2 (10.0) | 0 (0.0) | 1 (12.5) | 1 (16.7) | 6 (8.5) | 0.42 |
| | Hypertriglyceridemia | 0 (0.0) | 3 (30.0) | 4 (30.8) | 6 (75.0) | 6 (100.0) | 61 (85.9) | <0.00001 |
| | Hypercholesterolemia | 0 (0.0) | 0 (0.0) | 1 (7.7) | 3 (37.5) | 2 (33.3) | 45 (63.4) | <0.00001 |
| | Low HDL | 0 (0.0) | 0 (0.0) | 1 (7.7) | 1 (12.5) | 0 (0.0) | 27 (38.0) | 0.01 |
| | High LDL | 3 (100.0) | 7 (70.0) | 10 (76.9) | 5 (62.5) | 6 (100.0) | 63 (88.7) | 0.17 |

SHT= Systolic hypertension; DHT=Diastolic hypertension; FPG=Fasting Plasma Glucose.

(P-value<0.00001) from 0.0 and 0.0 to 92.2% and 61.8% respectively as the TG/HDL-c ratio increased. This trend was also evident in each of the three stages of adolescence. The prevalence of high LDL in mid- and late adolescents, relative to increase in TG/HDL-c ratio were not important. Table 4 and Fig 2 illustrate the ROC curves, Z-value and P-value of AUC, cut-off point, sensitivity, specificity, and distance on the ROC curve for the detection of non-lipid cardiometabolic variables such as MetS, SHT, DHT, dFPG, for each of the early, mid- and late adolescence. An inverse relationship was observed between the stages of adolescence and cutoff for MetS (7.92, 7.31 and 5.10 respectively), SHT (2.94, 1.76 and 1.14 respectively) and DHT (2.85, 3.25 and 8.37 respectively). However, the ability of TG/HDL-c to predict MetS was stronger in mid-adolescents with AUC (std. err) of 0.82 (0.07), Z value (P-value) of 4.44 (<0.00001), sensitivity of 0.71 and specificity

**Table 4. Sensitivity (Sens), specificity (Spec), and distance in the receiver operating characteristic (ROC) curve for TG/HDL-c cutoffs for MetS, SHT, DHT and dFPG among Nigerian adolescents.**

| Area Under the Curve (AUC) | | | | | | | | | | | | |
|---|---|---|---|---|---|---|---|---|---|---|---|---|
| Statistics for ROC | Metabolic syndrome (Yes = 1; No = 0) | | | Systolic hypertension (Yes = 1; No = 0) | | | Diastolic hypertension (Yes = 1; No = 0) | | | Diabetic Fasting Plasma Glucose (Yes = 1; No = 0) | | |
| | Adolescence | | | | | | | | | | | |
| | Early | Mid- | Late | Early | Mid- | Late | Early | Mid | Late | Early | Mid | Late |
| AUC (Std Error) | 0.87 (0.04) | 0.82 (0.07) | 0.75 (0.06) | 0.53 (0.05) | 0.51 (0.07) | 0.46 (0.04) | 0.45 (0.05) | 0.67 (0.12) | 0.36 (0.19) | 0.45 (0.08) | 0.44 (0.10) | 0.48 (0.05) |
| Z-value (P-value) | 9.92 (<0.00001) | 4.44 (<0.00001) | 4.36 (<0.00001) | 0.61 (0.27) | 0.11 (0.46) | −0.88 (0.81) | −1.07 (0.86) | 0.91 (0.18) | −0.75 (0.77) | −0.65 (0.74) | −0.57 (0.72) | −0.51 (0.70) |
| Cutoff | ≥7.92 | ≥7.31 | ≥5.10 | ≥2.94 | ≥1.76 | ≥1.14 | ≥2.85 | ≥3.25 | ≥8.37 | ≥0.72 | ≥5.0 | ≥0.55 |
| Sensitivity (95% CL) | 0.67 (0.43, 0.85) | 0.71 (0.29, 0.96) | 0.62 (0.41, 0.81) | 0.70 (0.53, 0.83) | 0.94 (0.71, 1.00) | 0.88 (0.77, 0.97) | 1.00 (0.16, 1.00) | 1.00 (0.16, 1.00) | 0.25 (0.01, 0.81) | 1.00 (0.79, 1.00) | 0.36 (0.11, 0.69) | 1.00 (0.92, 1.00) |
| Specificity (95% CL) | 0.90 (0.84, 0.94) | 0.85 (0.76, 0.91) | 0.81 (0.75, 0.85) | 0.44 (0.36, 0.52) | 0.30 (0.21, 0.40) | 0.15 (0.11, 0.20) | 0.41 (0.34, 0.48) | 0.49 (0.39, 0.58) | 0.91 (0.88, 0.94) | 0.04 (0.02, 0.08) | 0.70 (0.60, 0.79) | 0.04 (0.02, 0.08) |
| Distance to corner | 0.35 | 0.32 | 0.42 | 0.64 | 0.70 | 0.86 | 0.59 | 0.51 | 0.75 | 0.96 | 0.70 | 0.96 |

*Agex1 = Early adolescents (Blue line); Agex2 = Mid-adolescents (Green line); Agex3 = Late adolescents (Red line); AUC = Area Under Curve; ROC = Receiver Operating Character; *H0 (The hypothesis that Hypertension and TG/HDL-c are independent is not rejected).*

Area Under the Curve (AUC)

| Metabolic syndrome (Yes=1; No=0) | Systolic hypertension (Yes=1; No=0) | Diastolic hypertension (Yes=1; No=0) | Diabetic Fasting Plasma Glucose (Yes=1; No=0) |

**Fig 2. Sensitivity (Sens), specificity (Spec), and distance in the receiver operating characteristic (ROC) curve for TG/HDL-c cutoffs for MetS, SHT, DHT and dFPG among Nigerian adolescents.**

of 0.85, with the shortest distance to corner of 0.32. The TG/HDL-c ratio was unreliable in predicting systolic hypertension (cutoff ≥2.94, distance to corner = 0.64), diastolic hypertension (cutoff ≥2.85, distance to corner = 0.59), or diabetic fasting plasma glucose (cutoff ≥0.72, distance to corner = 0.96) in early adolescence or in any other stage of adolescence. Table 5 and Fig 3 illustrate the sensitivity, specificity, and distance in the receiver operating characteristic (ROC) curve for

TG/HDL-c cutoffs for dyslipidemia – hypertriglyceridemia, hypercholesterolemia, low high-density lipoprotein cholesterol and high low-density-lipoprotein cholesterol among the study subjects. TG/HDL-c ratio had a very strong ability to predict hypertriglyceridemia, especially among those in early adolescents, with an AUC (Std Err) of 0.91 (0.02), sensitivity of 0.86, specificity of 0.91, cutoff of ≥2.40 and distance to corner of just 0.17. The TG/HDL-c ratio was weaker in forecasting hypertriglyceridemia in late adolescents (cutoff=≥2.37, sensitivity=0.82, specificity=0.83, and distance to corner=0.24),

**Table 5. Sensitivity (Sens), specificity (Spec), and distance in the receiver operating characteristic (ROC) curve for TG/HDL-c cutoffs for hypertriglyceridemia, hypercholesterolemia, low high-density lipoprotein cholesterol and high low-density-lipoprotein cholesterol among Nigerian adolescents.**

| Statistics for ROC | Hypertriglyceridemia (Yes=1; No=0) | | | Hypercholesterolemia (Yes=1; No=0) | | | Low High-density lipoprotein cholesterol (Yes=1; No=0) | | | High Low-density lipoprotein cholesterol (Yes=1; No=0) | | |
|---|---|---|---|---|---|---|---|---|---|---|---|---|
| | Stages of adolescence | | | | | | | | | | | |
| | Early | Mid- | Late | Early | Mid- | Late | Early | Mid- | Late | Early | Mid- | Late |
| AUC (Std Error) | 0.91 (0.02) | 0.83 (0.04) | 0.90 (0.02) | 0.85 (0.03) | 0.87 (0.03) | 0.83 (0.02) | 0.10 (0.02) | 0.13 (0.04) | 0.17 (0.03) | 0.50 (0.07) | 0.54 (0.09) | 0.67 (0.05) |
| Z-value (P-value) | 15.94 (<0.00001) | 7.47 (<0.00001) | 19.84 (<0.00001) | 13.01 (<0.00001) | 11.00 (<0.00001) | 14.28 (<0.00001) | −16.47 (1.00) | −8.73 (1.00) | −10.40 (1.00) | 0.23 (0.49) | 0.45 (0.32) | 3.22 (0.0006) |
| Cut-off | ≥2.40 | ≥1.43 | ≥2.37 | ≥3.64 | ≥3.98 | ≥3.41 | ≥0.09 | ≥0.14 | ≥0.19 | ≥1.60 | ≥1.83 | ≥1.11 |
| Sensitivity (95% CL) | 0.86 (0.79, 0.91) | 0.95 (0.88, 0.99) | 0.82 (0.76, 0.87) | 0.84 (0.75, 0.90) | 0.76 (0.63, 0.87) | 0.73 (0.64, 0.81) | 1.00 (0.98, 1.00) | 1.00 (0.96, 1.00) | 1.00 (0.98, 1.00) | 0.79 (0.73, 0.85) | 0.76 (0.66, 0.84) | 0.91 (0.87, 0.95) |
| Specificity (95% CL) | 0.91 (0.80, 0.97) | 0.61 (0.42, 0.78) | 0.83 (0.74, 0.70) | 0.78 (0.69, 0.86) | 0.88 (0.77, 0.95) | 0.79 (0.72, 0.85) | 0.00 (0.00, 0.07) | 0.00 (0.00, 0.12) | 0.00 (0.00, 0.05) | 0.42 (0.25, 0.61) | 0.53 (0.28, 0.77) | 0.45 (0.29, 0.62) |
| Distance to corner | 0.17 | 0.39 | 0.24 | 0.27 | 0.26 | 0.34 | 1.00 | 1.00 | 1.00 | 0.62 | 0.53 | 0.56 |

*Agex1=Early adolescents (Blue line); Agex2=Mid-adolescents (Green line); Agex3=Late adolescents (Red line);* AUC=Area Under Curve; ROC=Receiver Operating Character; *H0 (The hypothesis that Hypertension and TG/HDL-c are independent is not rejected).

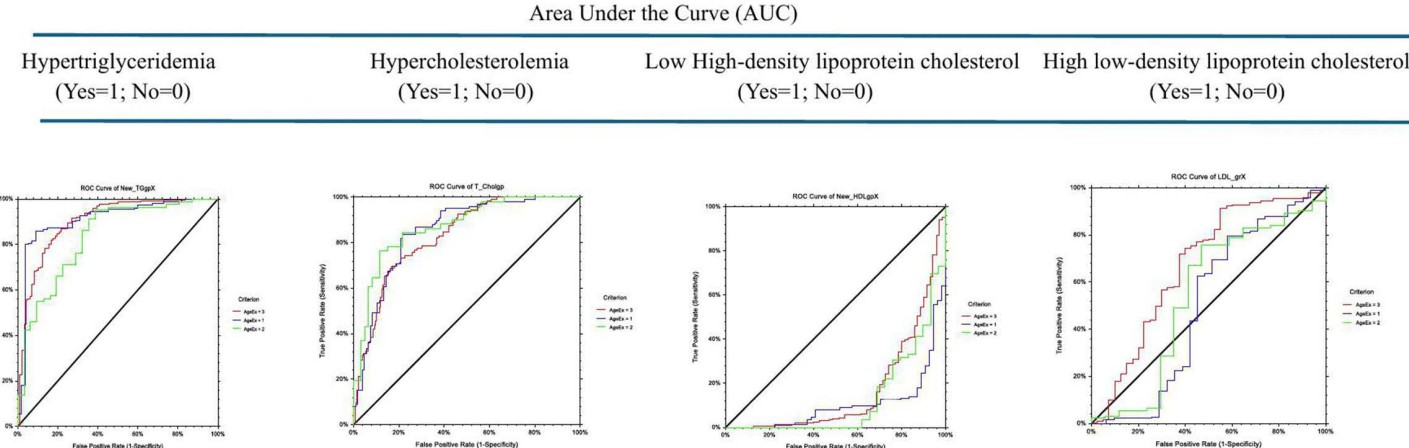

Area Under the Curve (AUC)

Hypertriglyceridemia (Yes=1; No=0)　Hypercholesterolemia (Yes=1; No=0)　Low High-density lipoprotein cholesterol (Yes=1; No=0)　High low-density lipoprotein cholesterol (Yes=1; No=0)

**Fig 3. Sensitivity (Sens), specificity (Spec), and distance in the receiver operating characteristic (ROC) curve for TG/HDL-c cutoffs for hypertriglyceridemia, hypercholesterolemia, low high-density lipoprotein cholesterol and high low-density-lipoprotein cholesterol among Nigerian adolescents.**

and weakest in predicting hypertriglyceridemia in mid-adolescents (distance to corner = 0.39). On the other hand, TG/HDL-c ratio was strong in predicting hypercholesterolemia both in early adolescence with an AUC (Std Err) of 0.85 (0.03), sensitivity of 0.84, specificity of 0.78, cutoff of ≥3.64 and distance to corner of 0.27 as well as in mid-adolescence, with an AUC (Std Err) of 0.87 (0.03), sensitivity of 0.76, specificity of 0.88, cutoff of ≥3.98 and distance to corner of 0.26. TG/HDL-c was practically inconsequential in predicting low HDL-c in any of the three stages of adolescence but slightly stronger in forecasting high LDL-c especially in mid adolescents with an AUC (Std Err) of 0.45 (0.09), sensitivity of 0.76, specificity of 0.53, cutoff of ≥1.83 and distance to corner of 0.53. Thus, the discriminatory power of TG/HDL-c ratio for cardiovascular risk factors among adolescent Nigerians was mainly limited to predicting MetS, hypertriglyceridemia, and hypercholesterolemia especially in early and in mid-adolescents. Table 6 and Fig 4 show the sensitivity, specificity, and distance in the

**Table 6. Sensitivity (Sens), specificity (Spec), and distance in the receiver operating characteristic (ROC) curve for TG/HDL-c cutoffs for dyslipidemia and risk factors of two or more among Nigerian adolescents.**

| Statistics for ROC | Dyslipidemia (Yes = 1; No = 0) | | | | Risk factors ≥2 (Yes = 1; No = 0) | | | |
|---|---|---|---|---|---|---|---|---|
| | Stages of adolescence | | | | | | | |
| | All | Early | Mid- | Late | All | Early | Mid- | Late |
| AUC (Std Error) | 0.87 (0.02) | 0.82 (0.03) | 0.88 (0.03) | 0.90 (0.02) | 0.81 (0.02) | 0.84 (0.03) | 0.77 (0.04) | 0.80 (0.03) |
| Z-value (P-value) | 23.58 (<0.00001) | 10.02 (<0.0001) | 11.85 (<0.00001) | 20.32 (<0.00001) | 17.44 (<0.00001) | 12.52 (<0.00001) | 5.92 (. <0.00001) | 11.83 (<0.00001) |
| Cut-off | ≥5.23 | ≥5.65 | ≥7.09 | ≥5.23 | ≥3.30 | ≥5.57 | ≥7.29 | ≥3.31 |
| Sensitivity (95% CL) | 0.94 (0.84, 0.99) | 0.95 (0.75, 1.00) | 1.00 (0.59, 1.00) | 0.92 (0.74, 0.99) | 0.76 (0.70, 0.81) | 0.67 (0.56, 0.77) | 0.41 (0.29, 0.56) | 0.72 (0.63, 0.79) |
| Specificity (95% CL) | 0.78 (0.75, 0.82) | 0.74 (0.64, 0.80) | 0.82 (0.73, 0.87) | 0.84 (0.79, 0.88) | 0.71 (0.66, 0.76) | 0.89 (0.83, 0.94) | 0.98 (0.91, 1.00) | 0.77 (0.70, 0.83) |
| Distance to corner | 0.22 | 0.26 | 0.18 | 0.18 | 0.37 | 0.35 | 0.59 | 0.37 |

*Agex1 = Early adolescents (Blue line); Agex2 = Mid-adolescents (Green line); Agex3 = Late adolescents (Red line); AUC = Area Under Curve; ROC = Receiver Operating Character; \*H0 (The hypothesis that Hypertension and TG/HDL-c are independent is not rejected).*

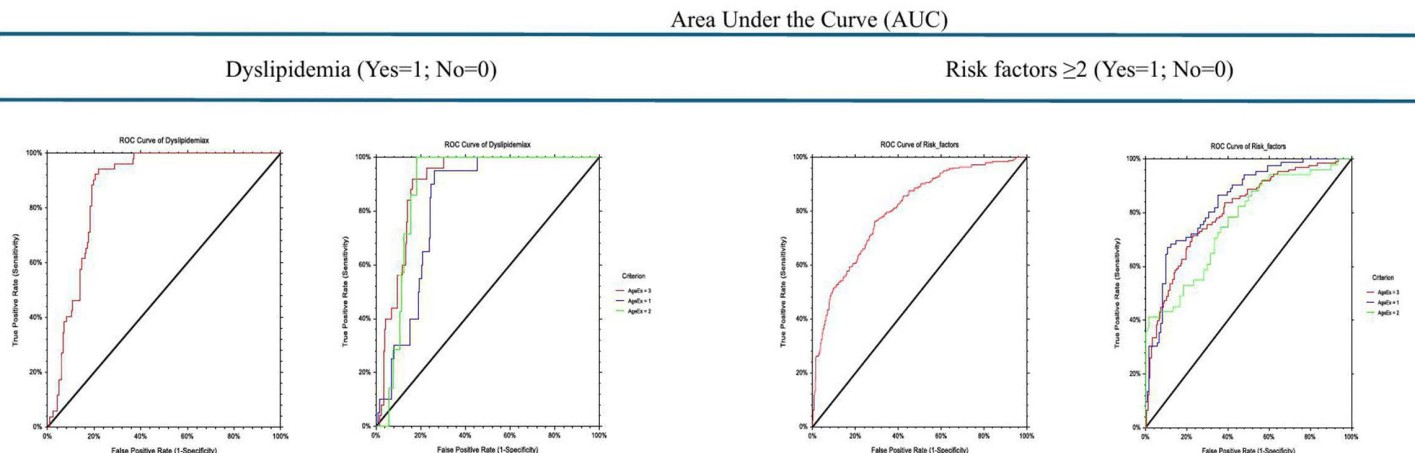

**Fig 4. Sensitivity (Sens), specificity (Spec), and distance in the receiver operating characteristic (ROC) curve for TG/HDL-c cutoffs for dyslipidemia and risk factors of two or more among Nigerian adolescents.**

receiver operating characteristic (ROC) curve for TG/HDL-c cutoffs for total dyslipidemia and risk factors of two or more among Nigerian adolescents. In all the study subjects, the shortest distance on the ROC curve for predicting dyslipidemia was 0.22, which lies between 0.18 (the shortest distance for predicting dyslipidemia in mid-adolescents and late adolescents respectively) and 0.26 (the shortest distance in early adolescents). The cutoff in early, mid- and late adolescents were 5.65, 7.09 and 5.23 respectively. The discriminatory power of TG/HDL-c ratio to predict ≥2 cardiovascular risk factors in all the study subjects was weaker, with the shortest distance on the ROC curve as 0.37, similar in late adolescents but lies between 0.35 in early and 0.59 in late adolescents. Table 7 illustrates the population distribution of each TG/HDL-c ratio and the sensitivity, specificity, and distance on the ROC curve for the detection of hypertension, dyslipidemia, diabetes, and ≥2 of these risk factors for all study subjects and for each of the three stages of adolescence, respectively.

## Discussion

This study provides age-specific-based data on the optimal cutoff of the triglyceride to HDL cholesterol (TG/HDL-c) ratio for detecting cardiovascular risk factors among a sub-Sahara African population in different stages of adolescence, aged between 10 and 19 years. To the best of the authors' knowledge, this may be the first cross-sectional study to evaluate the TG/HDL-c ratio cutoff among this group of people in this specific part of the globe. The study also reports an association between the TG/HDL-c ratio and cardiovascular risk factors. In sub-Saharan Africa, TG/HDL-c is a rather novel atherogenic tool for predicting MetS and other cardiometabolic variables, and the reports on this in the Nigerian adolescent population are rare. To detect MetS, the National Cholesterol Education Program (NCEP) Adult Treatment Panel III (ATPIII) standards registered optimal cutoff for TG/HDL-c as 1.6 and 1.2 in men and 1.1 in women, respectively [42]. Studies in China described the cutoff values of TG/HDL-c to identify MetS as 0.88 [43] and 0.84 in Zhuhai women [44]. In an African study, the cutoff value in Ghanaian women was reported as 0.61 [45] and in Canadian men and women, it was reported as 1.62 and 1.18, respectively [46]. In contrast to these reports, in the present study, consequent upon the sensitivity, specificity, and ROC calculations, the optimal cutoffs of TG/HDL-c ratio for Nigerian adolescents were 7.92 in early adolescence, 7.31 in mid adolescence and 5.10 in late adolescence respectively. The reasons for these high cutoff points among Nigerian adolescents are not understood. Future studies may be able to elucidate probable nutritional, dietary, environmental, climatic, living standard, genetic or even hereditary basis for the variation in these cutoffs in different population groups. At this point, it can only be speculated that these cutoffs are relatively high in adolescents because the risk factor for MetS and cardiovascular diseases are in the formative stages. In adolescence, TG,HDL-c has a weak predictive value for systolic and diastolic hypertension and failed to forecast diabetic fasting blood glucose in Nigerian adolescents. This may suggest that systolic hypertension, (and less so diastolic hypertension) may not necessarily be an integral aspect of MetS. Although studies have linked the occurrence of MetS to diabetes and insulin resistance [47–49], a study reported that TG/HDL-c ratio is not a reliable tool as marker of insulin resistance, especially in African Americans [50], which resonates with the findings of this study. It should be noted however, that a contrary view is held by Baneu et al, who stated that the predictive power of TG/HDL ratio varies across ethnicities and sexes, with specific thresholds providing greater accuracy for Caucasians, Asians, and Hispanics over African Americans and for men over women [51]. Evidence also abound of variations in racial and ethnic background of the actual values of the ratio that best identifies individuals [48,50,52]. It is also noted that variations in the optimal values of TG/HDL-c ratio between populations might be due to varied body size, physical activity, metabolic status [43] and age difference. Additionally, this study showed an increasing trend in the prevalence of total cholesterol and a decreasing trend in high-density lipoprotein with higher TG/HDL-c in all the adolescents when they were not segregated into the three stages. Both increasing and decreasing trends were observed in total cholesterol and high-density lipoprotein with increasing TG/HDL-c in early adolescents but not mid- or late adolescence. The reason for this is unclear but probably indicates the initial stage of atherogenic risk factors as basis for future variables such as hypertension and insulin resistance. Dharuni et al suggested that the elements that comprise metabolic syndrome are separately related with later higher risk for all-cause mortality, particularly cardiovascular disease

**Table 7. Sensitivity (Sens), specificity (Spec), and distance in the receiver operating characteristic (ROC) curve for each TG/HDL-c cutoff in adolescent Nigerians.**

| TG/HDL-c ratio | | Percentile | Diabetic FPG | | | Hypertriglyceridemia | | | Hypercholesterolemia | | | Low HDL-c | | | High LDL-c | | | ≥2 risk factors | | |
|---|---|---|---|---|---|---|---|---|---|---|---|---|---|---|---|---|---|---|---|---|
| | | | Sen (%) | Spec (%) | Dist. ROC curve | Sen (%) | Spec (%) | Dist. ROC curve | Sen (%) | Spec (%) | Dist. ROC curve | Sen (%) | Spec (%) | Dist. ROC curve | Sen (%) | Spec (%) | Dist. ROC curve | Sen (%) | Spec (%) | Dist. ROC curve |
| All | ≤1.0 | 2.6 | 100.0 | 0.0 | 0.93 | – | – | – | – | – | – | – | – | – | 100.0 | 0.0 | 1.00 | – | – | – |
| | >1.0 to ≤2.0 | 8.3 | 16.7 | 89.7 | 0.83 | 87.5 | 30.0 | 0.71 | – | – | – | 83.3 | 66.7 | 0.37 | 3.0 | 100.0 | 0.97 | 20.0 | 91.3 | 0.80 |
| | >2.0 to ≤3.0 | 12.2 | 66.7 | 42.0 | 0.66 | 52.6 | 91.1 | 0.48 | 40.0 | 98.6 | 0.60 | 31.5 | 100.0 | 0.68 | 54.8 | 61.5 | 0.59 | 28.6 | 89.7 | 0.72 |
| | >3.0 to ≤4.0 | 8.3 | 100.0 | 34.1 | 0.66 | 35.5 | 80.0 | 0.68 | 90.9 | 27.5 | 0.73 | 59.6 | 75.0 | 0.48 | 47.5 | 72.7 | 0.59 | 91.7 | 23.1 | 0.77 |
| | >4.0 to ≤5.0 | 7.8 | 100.0 | 28.6 | 0.71 | 80.0 | 55.6 | 0.49 | 87.5 | 53.1 | 0.49 | 51.2 | 80.0 | 0.53 | 69.6 | 100.0 | 0.30 | 38.5 | 80.0 | 0.65 |
| | >5.0 | 60.7 | 100.0 | 9.6 | 0.90 | 75.5 | 65.5 | 0.42 | 80.9 | 62.0 | 0.43 | 100.0 | 0.0 | 1.00 | 94.3 | 15.4 | 0.85 | 58.0 | 79.7 | 0.47 |
| Early adolescence | ≤1.0 | 3.0 | – | – | – | – | – | – | – | – | – | – | – | – | 16.7 | 100.0 | 0.83 | – | – | – |
| | >1.0 to ≤2.0 | 8.4 | 100.0 | 11.1 | 0.89 | 100.0 | 34.8 | 0.65 | – | – | – | 81.8 | 66.7 | 0.40 | 23.1 | 83.3 | 0.79 | 33.3 | 90.9 | 0.67 |
| | >2.0 to ≤3.0 | 11.8 | 100.0 | 3.2 | 0.97 | 57.1 | 89.3 | 0.44 | 100.0 | 0.0 | 0.0 | 47.1 | 100.0 | 0.53 | 90.3 | 75.0 | 0.27 | 75.0 | 51.6 | 0.54 |
| | >3.0 to ≤4.0 | 9.8 | 25.0 | 100.0 | 0.75 | 52.9 | 83.3 | 0.50 | 66.7 | 56.5 | 0.55 | 40.7 | 100.0 | 0.59 | 30.4 | 83.3 | 0.71 | 57.1 | 63.6 | 0.56 |
| | >4.0 to ≤5.0 | 9.4 | 60.0 | 65.2 | 0.53 | 100.0 | 30.0 | 0.70 | 90.0 | 50.0 | 0.51 | 96.0 | 33.3 | 0.67 | – | – | – | 100.0 | 18.2 | 0.82 |
| | >5.0 | 57.6 | 100.0 | 12.2 | 0.88 | 71.3 | 71.4 | 0.40 | 76.8 | 67.1 | 0.40 | 100.0 | 0.0 | 1.00 | 93.6 | 20.0 | 0.80 | 53.4 | 80.9 | 0.50 |
| Mid adolescence | ≤1.0 | 25.0 | – | 0.0 | – | – | – | – | – | – | – | – | – | – | 100.0 | 0.0 | 1.00 | – | – | – |
| | >1.0 to ≤2.0 | 31.4 | 33.3 | 100.0 | 0.67 | 100.0 | 38.5 | 0.62 | – | – | – | – | – | – | 15.4 | 100.0 | 0.85 | – | – | – |
| | >2.0 to ≤3.0 | 36.0 | 100.0 | 36.0 | 0.64 | 37.5 | 94.7 | 0.63 | 33.3 | 100.0 | 0.67 | – | – | – | 61.9 | 66.7 | 0.51 | 100.0 | 36.0 | 0.64 |
| | >3.0 to ≤4.0 | 27.5 | 100.0 | 58.3 | 0.42 | 100.0 | 33.3 | 0.67 | 100.0 | 25.0 | 0.75 | 46.1 | 100.0 | 0.54 | 100.0 | 100.0 | 0.00 | 100.0 | 41.7 | 0.58 |
| | >4.0 to ≤5.0 | 29.2 | – | – | – | 66.7 | 100.0 | 0.33 | 100.0 | 70.0 | 0.30 | 41.7 | 100.0 | 0.58 | 58.3 | 100.0 | 0.42 | 75.0 | 70.0 | 0.39 |
| | >5.0 | 34.9 | 88.9 | 27.3 | 0.74 | 96.0 | 60.0 | 0.40 | 90.0 | 45.0 | 0.56 | 100.0 | 0.0 | 1.00 | 95.6 | 12.5 | 0.88 | 74.3 | 76.8 | 0.35 |
| Late adolescence | ≤1.0 | 18.7 | 100.0 | 0.0 | 1.0 | – | – | – | – | – | – | – | – | – | – | – | – | – | – | – |
| | >1.0 to ≤2.0 | 19.6 | 50.0 | 75.0 | 0.56 | 100.0 | 14.3 | 0.86 | – | 58.3 | 0.42 | – | – | – | 71.4 | 33.3 | 0.73 | 100.0 | 12.5 | 0.87 |
| | >2.0 to ≤3.0 | 17.3 | – | – | – | 100.0 | 77.8 | 0.22 | 100.0 | 100.0 | 0.67 | 25.0 | 100.0 | 0.75 | 40.0 | 100.0 | 0.60 | 100.0 | 8.3 | 0.92 |
| | >3.0 to ≤4.0 | 15.7 | 100.0 | 57.1 | 0.43 | 100.0 | 50.0 | 0.50 | 33.3 | 100.0 | 0.50 | 85.7 | 100.0 | 0.14 | 40.0 | 100.0 | 0.60 | 66.7 | 60.0 | 0.52 |
| | >4.0 to ≤5.0 | 12.5 | 100.0 | 60.0 | 0.40 | – | – | – | 50.0 | 100.0 | 0.50 | – | – | – | – | – | – | 66.7 | 55.7 | 0.47 |
| | >5.0 | 19.1 | 66.7 | 53.9 | 0.57 | 55.7 | 90.0 | 0.45 | 86.7 | 73.1 | 0.30 | 100.0 | 0.0 | 1.00 | 96.8 | 12.5 | 0.88 | 50.0 | 96.5 | 0.50 |

[53]. Nonetheless, this study evaluated TG/HDL-c ratio in the framework of cardiometabolic risk expectation. Thus, data from this study shows that the TG/HDL-c ratio is a meaningful predictor of cardiometabolic risk factors in this adolescent population. The inverse relationship between stages of adolescence and TG/HDL-c cutoff for predicting MetS suggests a gradual development of cardiometabolic component of MetS which may be rarer in early but more frequent in late adolescents. The observed correlations between higher TG/HDL-c ratios and increased markers of individual cardiometabolic risk, such as hypertriglyceridemia, hypercholesterolemia emphasize the significance of this ratio as a principal indicator of potential health risks in adolescence. Studies have shown that hypertriglyceridemia is associated with a prolonged risk of atherosclerotic cardiovascular disease (ASCVD) [54]. Unchecked mild-to moderate hypertriglyceridemia of 150–499 mg/dL concentration could lead to severe form of >880 mg/dL and, at a higher concentration of >1770mg/dL, to acute pancreatitis, a clinically important risk [55] and multiorgan failure [56]. In this study, the median value of TG/HDL-c ratio among adolescents with hypertriglyceridemia (4.2) was significantly higher (P-value<0.00001) than that among those with normal triglyceride concentration (1.3). This is similar to the findings of Dharuni et al in in an adult population in South India [53] which reported that TG/HDL-c ratio is not only a strong independent predictor of myocardial infarction (MI) but also that of coronary heart disease (CHD). Da Luz et al, in a Brazilian study, reported that TG/HDL-c ratio exhibited the strongest link with coronary disease scored by Friesinger index [57] while Shou P et al, in a Chinese study on adults, showed that TG/HDL-c was the best prognostic factor of MetS in women whereas in men waist-hip ratio was the best predictor [58]. According to a Copenhagen Male Study, TG/HDL-c ratios can more precisely identify and distinguish coronary disease [59]. Edwards et al [60] and Brinton and his colleagues [61] suggested that the conceivable pathway is that high triglycerides and low HDL-c form the foundation for the buildup of small and dense LDL-c, causing HDL-c particles to initiate accelerated catabolism, thus concluding the atherogenic circle. The current study suggests that the cutoff of TG/HDL-c ratio is a good predictor of cardiometabolic risk factors, especially dyslipidemia but not systolic hypertension or diabetic fasting plasma glucose. With its high sensitivity and specificity, and it will be about the most appropriate means of predicting the incidence of current (in adolescence) and possible future (in adulthood) atherosclerotic diseases in Nigeria.

## Strengths and limitations

This study specified facts and figures on cutoffs and AUC for specific atherogenic variables stratified by three stages of adolescence – early, mid and late. Other researchers, especially in sub-Saharan Black Africa can utilize the cutoffs proposed in this study for the screening and clinical intervention of cardiovascular disease especially in this age group. However, the study has some limitations that need further consideration. First, the sample size is relatively small. A larger sample size would probably have given a more robust analysis and extend its external validity. Second, bias may have been introduced in the sample selection as the proportion of girls outweighed that of boys, and some of the overweight or obese students may have been left out. Thirdly, the study was conducted in the southwest corner of Nigeria, on the Atlantic Ocean coastline thus it may not be representative of Nigerian adolescent population in the arid eco-system, the population living in high altitude region in north-central part of the country or those resident in the rain-forest zone. Fourthly, this was a cross-sectional study thus devoid of predicting any prognosis nor does it assume cause and effect of any findings. Finally, the study did not consider anthropometric status such as waist circumference, Body Mass index-for-age to determine lean, overweight or obese adolescents, sex, or ethnicity. It is expected that future studies will be able to recruit a much larger number of participants, probably in a national or multi-center study of TG/HDL-c.

## Conclusions

To conclude, this study showed a TG/HDL-c ratio effectively predicted MetS in early, mid- and late adolescent stages with cutoff values of 7.92, 7.31 and 5.10; hypertriglyceridemia with cutoff of 2.40, 1.43 and 2.37 respectively; and hypercholesterolemia with cutoff of 3.64, 3.98 and 3.41 respectively to differentiate high cardiovascular risk adolescents. The study also documented the continuous relationship between two cardiovascular disease risk factors – total cholesterol and

high-density lipoprotein in all the study subjects and TG/HDL-c ratio. As a tool, the TG/HDL-c ratio should be considered for the initial prediction of MetS and dyslipidemia.

## Acknowledgments

Dr. Bamgboye M. Afolabi is the guarantor of this work and, as such, has full access to all the data in the study and takes responsibility for the integrity of the data and the accuracy of the data analysis. This project was supported by the Nigerian Institute of Medical Research (NIMR). We would like to thank the Lagos State Ministries of Education, Health and Public Service and their districts heads for making this study possible. We also thank the parents of the students as well as the students for making this study possible.

## Author contributions

**Conceptualization:** Susan J. Holdbrooke, Bamgboye M. Afolabi.

**Data curation:** Susan J. Holdbrooke, Bamgboye M. Afolabi.

**Formal analysis:** Susan J. Holdbrooke, Bamgboye M. Afolabi.

**Investigation:** Bamgboye M. Afolabi.

**Methodology:** Bamgboye M. Afolabi.

**Project administration:** Susan J. Holdbrooke.

**Resources:** Susan J. Holdbrooke.

**Software:** Bamgboye M. Afolabi.

**Validation:** Bamgboye M. Afolabi.

**Writing – original draft:** Bamgboye M. Afolabi.

**Writing – review & editing:** Bamgboye M. Afolabi.

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
