## [Decision Letter · Decision Letter 0]

20 May 2025

Dear Dr. Afolabi,

Thank you for submitting your manuscript to PLOS ONE. After careful consideration, we feel that it has merit but does not fully meet PLOS ONE’s publication criteria as it currently stands. Therefore, we invite you to submit a revised version of the manuscript that addresses the points raised during the review process.

We look forward to receiving your revised manuscript.

Kind regards,

Paolo Magni

Academic Editor

PLOS ONE

Journal Requirements:

Additional Editor Comments:

The paper has been improved. We ask you to further address in full the issues raised by Reviewer 1.

Reviewers' comments:

Reviewer's Responses to Questions

**Comments to the Author**

1. Is the manuscript technically sound, and do the data support the conclusions?

Reviewer #1: Yes

Reviewer #2: Yes

2. Has the statistical analysis been performed appropriately and rigorously?

Reviewer #1: Yes

Reviewer #2: Yes

3. Have the authors made all data underlying the findings in their manuscript fully available?

Reviewer #1: Yes

Reviewer #2: Yes

4. Is the manuscript presented in an intelligible fashion and written in standard English?

Reviewer #1: Yes

Reviewer #2: Yes

Reviewer #1: It is a very well written article regarding Predictive Ability Of Plasma Concentration Of Triglyceride/High Density Lipoprotein-Cholesterol Ratio For Cardiometabolic variables In A Sub-Sahara Black African Adolescent Population – Nigerians.

I want to congratulate the authors for their thorough effort.

I have only 2 comments to make.

1. In the Introduction part, they should include a more detailed discussion regarding the attributes and clinical usefulness of the TG/HDL ratio. A pertinent review article referring to the subject is:

Kosmas CE, Rodriguez Polanco S, Bousvarou MD, Papakonstantinou EJ, Peña Genao E, Guzman E, Kostara CE. The Triglyceride/High-Density Lipoprotein Cholesterol (TG/HDL-C) Ratio as a Risk Marker for Metabolic Syndrome and Cardiovascular Disease. Diagnostics (Basel). 2023 Mar 1;13(5):929. doi: 10.3390/diagnostics13050929. PMID: 36900073; PMCID: PMC10001260.

I feel that a brief discussion of the above review article and its addition to the references would be useful.

2. Since the authors appear to have the data of fasting plasma glucose of the population of their study, it would be very useful and would add to the complicity to the study if they would also examine the validity of the triglyceride glucose (TyG) index (another very well established index) on the same cardiometabolic variables and possibly compare the predictability of the TyG index with the predictability of the TG/HDL ratio. TyG index = Ln [TG (mg/dL) × fasting glucose (mg/dL)/2]. A pertinent article referring to this is:

Song K, Park G, Lee HS, Choi Y, Oh JS, Choi HS, Suh J, Kwon A, Kim HS, Chae HW. Prediction of Insulin Resistance by Modified Triglyceride Glucose Indices in Youth. Life (Basel). 2021 Mar 28;11(4):286. doi: 10.3390/life11040286. PMID: 33800541; PMCID: PMC8066260.

I feel that a brief discussion of the above review article and its addition to the references would also be useful.

Reviewer #2: This is an automated report for PONE-D-25-13856. This report was solicited by the PLOS One editorial team and provided by ScreenIT.

ScreenIT is an independent group of scientists developing automated tools that analyze academic papers. A set of automated tools screened your submitted manuscript and provided the report below. Each tool was created by your academic colleagues with the goal of helping authors. The tools look for factors that are important for transparency, rigor and reproducibility, and we hope that the report might help you to improve reporting in your manuscript. Within the report you will find links to more information about the items that the tools check. These links include helpful papers, websites, or videos that explain why the item is important. While our screening tools aim to improve and maintain quality standards they may, on occasion, miss nuances specific to your study type or flag something incorrectly. Each tool has limitations that are described on the ScreenIT website. The tools screen the main file for the paper; they are not able to screen supplements stored in separate files. Please note that the Academic Editor had access to these comments while making a decision on your manuscript. The Academic Editor may ask that issues flagged in this report be addressed. If you would like to provide feedback on the ScreenIT tool, please email the team at ScreenIt@bih-charite.de. If you have questions or concerns about the review process, please contact the PLOS One office at plosone@plos.org.

**Do you want your identity to be public for this peer review?** For information about this choice, including consent withdrawal, please see our Privacy Policy

Reviewer #1: No

Reviewer #2: No

---

## [Author Response · Author response to Decision Letter 1]

30 May 2025

My collegue and I would like to express our gratitute to you and the two reviewers who had gone through our original manuscript titled “Predictive Ability Of Plasma Concentration Of Triglyceride/High Density Lipoprotein-Cholesterol Ratio For Cardiometabolic variables In A Sub-Sahara Black African Adolescent Population – Nigerians.” We appreciate the efforts of everyone to make this manuscript as welcoming and enjoyable to read, as much as possible. We have followed the Editor’s remarks one by one and responded to them. All the remarks and corrections suggested by the reviewers have been effected.

Rebutal

However, we would like to crave your indulgence to defer the inclusion of TgG ratio to a future paper. The main reason for this deferment is that there are already, too many information included in the current manuscript and we would like to avoid readers’ fatigue. Another reason is that, including TgG ratio will cause a complete overall of the current paper. For these reasons, we wish to kindly say that we would consider a separate manuscript in the future to evaluate TgG ratio as a valid instrument of determining metabolic syndrome among Nigerian adolescents.

We are also happy to inform your good offices that we have included a flowchart into the paper as suggested.

Response to Editor and Reviewers

We hereby confirm that our submission contains all raw data required to replicate the results of our study.

Reviewer 1’s comment

1. In the Introduction part, they should include a more detailed discussion regarding the attributes and clinical usefulness of the TG/HDL ratio. A pertinent review article referring to the subject is: Kosmas CE, Rodriguez Polanco S, Bousvarou MD, Papakonstantinou EJ, Peña Genao E, Guzman E, Kostara CE. The Triglyceride/High-Density Lipoprotein Cholesterol (TG/HDL-C) Ratio as a Risk Marker for Metabolic Syndrome and Cardiovascular Disease. Diagnostics (Basel). 2023 Mar 1;13(5):929. doi: 10.3390/diagnostics13050929. PMID: 36900073; PMCID: PMC10001260.I feel that a brief discussion of the above review article and its addition to the references would be useful.

Our response: We have included and highlighted a brief discussion regarding the attributes and clinical usefulness of the TG/HDL-C ratio in the Introduction segment of the manuscript.

Reviewer 2’s commets using ScreenIT software:

Flow charts and attrition

We did not find a study flow chart of excluded observations. We strongly recommend using flow charts because they provide an overview of the study design and more information about attrition. If you included a study flow chart in your supplemental files, we apologize for missing this. Our tool is not able to screen separate supplemental files.

Sentence about attrition: not detected. Please provide information about the drop-out of subjects, or loss of animals or samples. This could be done using a study flow chart, or described in the text.

Our response: Flow chart and attrition have been included in the manuscript.

Randomization

Not detected. If you performed an experimental study, please specify whether participants, animals or samples were randomly assigned to treatments or groups, and specify the randomization procedure.

Our response: We did not perform any experimental study.

Blinding

Not detected. Please specify whether blinding was used at various phases of the experiment (e.g., blinding of patients, caregivers, outcome assessments, data analysis).

Our response: Blinding was not used in any stage of the study. The study was not experimental.

Transparency

If you wrote code to analyze your data, please consider sharing the code in a public repository. This makes it easier for others to reproduce your analyses, and may also aid others seeking to analyze similar datasets.

If permitted, sharing data on a public repository with a persistent identifier (a DOI or accession number) can improve reproducibility and make it easier for other scientists to expand on your work. Papers with open data are cited more often than papers without open data. Some institutions have an expert who can provide advice on data sharing.

Response: We did not use code in our data analysis.

---

## [Decision Letter · Decision Letter 1]

18 Jun 2025

Predictive Ability Of Plasma Concentration Of Triglyceride/High Density Lipoprotein-Cholesterol Ratio For Cardiometabolic variables In A Sub-Sahara Black African Adolescent Population – Nigerians.

PONE-D-25-13856R1

Dear Dr. Afolabi,

<table border="0" cellpadding="0" cellspacing="0" class="datatable3"> <tbody> <tr> <td style="width:30%">To:</td> <td style="text-align:left;">Bamgboye M. Afolabi</td> </tr> </tbody></table><table border="0" cellpadding="0" cellspacing="0" class="datatable3"> <tbody> <tr> <td style="width:30%">To:</td> <td style="text-align:left;">Bamgboye M. Afolabi</td> </tr> </tbody></table>

We’re pleased to inform you that your manuscript has been judged scientifically suitable for publication and will be formally accepted for publication once it meets all outstanding technical requirements.

Kind regards,

Paolo Magni

Academic Editor

PLOS ONE

Additional Editor Comments (optional):

All reviewers' comments have been addressed.

Reviewers' comments:

Reviewer's Responses to Questions

**Comments to the Author**

Reviewer #1: All comments have been addressed

2. Is the manuscript technically sound, and do the data support the conclusions?

Reviewer #1: Yes

3. Has the statistical analysis been performed appropriately and rigorously?

Reviewer #1: Yes

4. Have the authors made all data underlying the findings in their manuscript fully available?

Reviewer #1: Yes

5. Is the manuscript presented in an intelligible fashion and written in standard English?

Reviewer #1: Yes

Reviewer #1: The authors have adequately addressed the comments raised in the previous round of review and I feel that this manuscript is now acceptable for publication.

**Do you want your identity to be public for this peer review?** For information about this choice, including consent withdrawal, please see our Privacy Policy

Reviewer #1: No

---

## [Editor Report · Acceptance letter]

PONE-D-25-13856R1

PLOS ONE

Dear Dr. Afolabi,

I'm pleased to inform you that your manuscript has been deemed suitable for publication in PLOS ONE. Congratulations! Your manuscript is now being handed over to our production team.

Kind regards,

on behalf of

Prof. Paolo Magni

Academic Editor

PLOS ONE